# Population’s Potential Accessibility to Specialized Palliative Care Services: A Comparative Study in Three European Countries

**DOI:** 10.3390/ijerph181910345

**Published:** 2021-09-30

**Authors:** Danny van Steijn, Juan José Pons Izquierdo, Eduardo Garralda Domezain, Miguel Antonio Sánchez-Cárdenas, Carlos Centeno Cortés

**Affiliations:** 1ATLANTES Research Group, Institute for Culture and Society, University of Navarra, 31009 Pamplona, Navarra, Spain; jpons@unav.es (J.J.P.I.); egarralda@unav.es (E.G.D.); sanchezcmiguel@unav.es (M.A.S.-C.); ccenteno@unav.es (C.C.C.); 2Navarra Institute for Health Research (IdiSNA), Recinto de Complejo Hospitalario de Navarra C/Irunlarrea, 3, 31008 Pamplona, Navarra, Spain; 3School of Humanities and Social Sciences, University of Navarra, 31009 Pamplona, Navarra, Spain

**Keywords:** palliative care, geographical accessibility, equity, Europe

## Abstract

Background: Palliative care is a priority for health systems worldwide, yet equity in access remains unknown. To shed light on this issue, this study compares populations’ driving time to specialized palliative care services in three countries: Ireland, Spain, and Switzerland. Methods: Network analysis of the population’s driving time to services according to geolocated palliative care services using Geographical Information System (GIS). Percentage of the population living within a 30-min driving time, between 30 and 60 minutes, and over 60 min were calculated. Results: The percentage of the population living less than thirty minutes away from the nearest palliative care provider varies among Ireland (84%), Spain (79%), and Switzerland (95%). Percentages of the population over an hour away from services were 1.87% in Spain, 0.58% in Ireland, and 0.51% in Switzerland. Conclusion: Inequities in access to specialized palliative care are noticeable amongst countries, with implications also at the sub-national level.

## 1. Introduction

Palliative care improves the quality of life of patients and that of their families who are facing challenges associated with life-threatening illness, whether physical, psychological, social, or spiritual [1]. In Europe, 4.4 million people died in need of palliative care in 2014, 139,000 of whom were children [2]. The future is even more concerning as projections estimate that the global need will continue to grow [3], and therefore, access to palliative care becomes an urgent and important global health matter. Accordingly, the World Health Organization (WHO) identifies Universal Health Coverage (access for everyone to the health-care services they need) as a strategic priority [4] and in 2014 passed a global resolution that urged member states to include palliative care in the definition of Universal Health Coverage (WHA 67.19 [5]). International consensus holds that palliative care networks can relieve the burden on health systems by allowing patients to remain at home or in communities, thus reducing hospital admissions [6].

Access to palliative care may be approached differently, but certainly, the geographical approach is crucial and contributes to drawing the equity or inequity of access, which is particularly relevant where inequities are presumable such as in in Low- and Middle-Income Countries [7,8,9,10]. The WHO has repeatedly named it as an important indicator [11] and as a key characteristic of a good service delivery [12]. Subsequently, an overview of the geographic accessibility of a country to specialized services is necessary for a good evidence-based spatial planning.

The concept of accessibility to health services has been explored as a part of access, alongside availability, accommodation, affordability, and acceptability [13]. Geographical or spatial accessibility to health care has been defined as the relationship between the location of supply and the location of clients, taking account of client transportation resources and travel time, distance, and cost [13,14,15]. Two types of dichotomies have been used for accessibility: potential versus revealed accessibility or utilization [16], and spatial (geographic) and aspatial (social) access [17]. Research has explored different factors of accessibility such as the land use, transportation, temporal, and individual factors [18]. Individual factors are more used by qualitative research including topics such as the perception of (in)acceptability of palliative care services [19], community readiness [20], and patients’ perspectives on health care in rural communities [21]. Others have explored the individual constraints for accessibility, such as not being able to afford to travel or the difficulty of riding long distances with a broken arm [21], distrust in hospitals [19], and car ownership or health status [22,23].

The majority of the literature on geographical access to palliative care has focused on land use and/or transportation components using Geographical Information Systems (GIS). These studies generally use a distance measure to calculate the population accessibility to health-care services. Most studies have used a regional scale [20,23,24,25,26,27,28], others used a national scale [29,30,31,32,33,34], and few employed an urban scale [19,35]. These included several types of palliative care services (hospital resources, inpatient units, (pediatric) hospices, home-care services, etc.). Other than a study comparing England and Wales [33], no other study has compared countries.

Comparative methodology is a universally accepted research method. A comparative approach can test assumptions about how well a system, a policy, or a procedure works in different contexts. Furthermore, comparing countries can suggest new ideas and provide good reference examples at the time that increases explanatory power for inductive reasoning (would a successful policy in one country work in another?) and deductive explanation (are there exceptions to a generalization?) [36]. Regarding accessibility, comparing countries might show similar or differing factors underlying accessibility and can serve as a reference to other countries on how to improve their accessibility to specialized palliative care services. In this sense, a replicable method to estimate the population’s access to palliative care services in several countries would be useful for evidence-based health planning.

Due to a lack of data on the actual use of palliative care services that would permit a revealed accessibility study, the aim of this paper is to estimate the population’s potential accessibility to specialized palliative care services in three countries: Ireland, Spain, and Switzerland.

## 2. Materials and Methods

This study adapts the method of Schuurman et al. [37], using a rational catchment method to create service areas around specialized palliative care services, and later defining the population living within service areas of 30-minute intervals (0–30, 30–60, and over 60). The choice of these time limits has been established following relevant articles on geographic accessibility to palliative care services [38,39], as per considering the special fragility of a person with palliative care needs and their priority in access.

### 2.1. Study Setting

Ireland, Spain, and Switzerland were chosen based on diverse extension, population, and relief; and on the availability for the three countries of a National Palliative Care Directory with postal addresses that permitted the geolocation of the specialized palliative care services.

The Republic of Ireland occupies the biggest part (70,280 km^2^) of the Island of Ireland. It has a population of more than 4.9 million people, which means a density of 71 inhabitants/km^2^. Many people live in and around urban centers such as Dublin and Cork, but with 37% rural population [40], Ireland has the highest proportion of rural population among the three countries. Ireland is relatively flat in the center but has some hill ranges in the north and south, along the western coast, and in the Wicklow Mountains at the south of Dublin. From an administrative viewpoint, the local government is divided into 26 county councils, three city councils and two city–county councils. The smallest administrative areas are the electoral divisions, of which there are 3409.

Spain is the largest and most populous country in this study. With more than half a million square kilometers area and a population of 47.3 million people, it has an average density of 93.5 inhabitants/km^2^. The Spanish population is very unevenly distributed in the territory. There are big urban areas such as Madrid or Barcelona, with 5300 inhabitants/km^2^ and 16,000 inhabitants/km^2^, and wide interior regions with less than five inhabitants/km^2^. The rural population of Spain represents 19% of the country’s population. Spain’s orography is characterized by many mountain ranges: The Pyrenees on its northern border, the Sistema Central at the Northwest of Madrid, the Sistema Ibérico at the Northeast of Madrid, and the Baetic System in Southern Spain. Spain is a decentralized political system divided into 17 autonomous regions and two autonomous cities, some of which are further divided into several provinces. The lowest level administrative units are the municipalities. As of now, there exist 8131 municipalities.

The landlocked country of Switzerland is the smallest country in area of this study, with a surface of 41,290 km^2^ and a population of 8.5 million people. It has the highest population density of studied areas: 215 inhabitants/km^2^. With its location partly in the Alps, Switzerland has many mountainous areas. The country has a rural population of 26%. Switzerland is a decentralized political system divided into 26 cantons. More than half the population live in only four of them. Switzerland consists of 2215 municipalities.

### 2.2. Specialized Palliative Care Services

Data on specialized palliative care services were extracted in late 2019 from the national palliative care directory websites of the respective countries [41,42,43]. The total number of specialized palliative care services and the services per population according to the national directories were 61 services in Ireland (1.2 per 100,000 inhabitants); 201 services in Spain (0.4 per 100,000 inhabitants); and 95 services in Switzerland. (1.1 per 100,000 inhabitants). All types of specialized palliative care services that comply with the directories’ criteria were included.

### 2.3. Service Areas

Services were geo-located based on their postal address in ArcGIS software. Service areas were made around them with the Network Analysis tool in ArcGIS Pro. The network dataset used for this (roads and their characteristics, such as speed, traffic lights, and signs) were gathered from the ArcGIS Online Cloud service, which makes use of data from HERE technologies (a proprietary dataset used for routing in turn-by-turn navigation systems [44]). According to the ESRI, the countries in this study have comprehensive street data with historical and live traffic data available [45]. The settings were set to ‘driving time in minutes toward specialized palliative care services’, and the cut-off times were 10, 20, 30, 40, 50, 60, and 90 min. No time of day was chosen, and the output geometry was adjusted to standard precision, dissolve, and rings. This created a polygon layer with the service areas based on the cut-off times. A merge was done between service areas to create the areas of 0–30, 30–60, and over 60, which is in accordance with other studies that have used 30-min [33,46] or 60-min driving time [37], as these were considered reasonably differentiated travel times. Given the high level of suffering of patients, distances in driving time from 30 min onwards were considered unreasonable [33,47,48].

### 2.4. Population within Service Areas

The demographic data were obtained from national statistics and geographic institutes: electoral divisions of Ireland [49,50], municipalities for Spain [51,52], and Switzerland [53,54]. To accurately mark the residential areas within service areas, and to lower a false assumption on a homogeneous distribution of the population, a dasymetric mapping method was applied [55]. Imperviousness density data were used as ancillary information [56]. These data contain a raster map of the level of imperviousness with a resolution of 20 m, showing built-up areas, which are used as a proxy for residential areas. The imperviousness layer was intersected with the small area layer, resulting in a layer that features the approximate residential areas containing the total population of the area. Next, the residential areas were intersected with the service areas layer. This was applied to calculate the population within each service area.

### 2.5. Data Presentation

Two main data will be shown: (1) surface coverage of services, and (2) potential population´s accessibility to services across countries and regions.

## 3. Results

### 3.1. Surface Coverage of Specialized Palliative Care Services

Notable differences exist in the surface coverage of specialized services amongst countries. In Ireland, less than 10% of the land stands over a 60-min drive from a specialized palliative care service, while 54% of the Irish surface has a specialized service in less than 30 min driving time. In Spain, 34% of the country’s territory is over a one-hour drive away from the nearest specialized palliative care service. Furthermore, just 20% of Spanish territories have a service in less than 30 min driving time, normally in the capitals of the provinces and surrounding areas. In Switzerland, more than 40% of the surface is over 60 min away from the nearest specialized service (Figure 1).

In Ireland, areas away from services are hilly coastal sites placed on the western seaside. In Spain, isolated areas correspond to the Pyrenees, the provinces of Soria, Palencia, and Zamora (Castilla y León), Teruel (Aragón), Guadalajara y Cuenca (Castilla-La Mancha), Orense (Galicia), southern parts of Ciudad Real and Albacete (Castilla-La Mancha), the western part of Córdoba (Andalucía), and the Canary Islands. While in northern Switzerland, nearly all the surface is less than a 30-min drive, the southern cantons of Valais, Ticino (except for cities such as Bellinzona or Lugano), and Grisons are over an hour driving time from services (Figure 2).

### 3.2. Potential Population’s Accessibility to Specialized Palliative Care Services

In all three countries, the majority of the population live within a 0 to 30 min driving time range to a specialized palliative care service (Ireland 84%, Spain 79%, and Switzerland 95%). The percentage of population living within 30–60 driving time range to a specialized palliative care service tend to be smaller, and the percentage of those living further than 60 min driving time are even smaller: 20.19% of Spaniards, 15.31% of Irish, and 4.81% of Swiss people live further than 30 min away from the nearest specialized service, of which 3.19%, 1.31%, and 0.81%, respectively, are over an hour drive. In absolute numbers, the total number of people living over a 30-min distance is 9,695,273 people in Spain, 741,950 people in Ireland, and 452,659 in Switzerland (Table 1).

Geographical accessibility to services in the three countries presents, besides many inequities across countries, big differences depending on the region in which the population lives. In Ireland, most of the regions with over 40% of people living further than 30 min of driving are along the north: Monaghan (92%), Leitrim (47%), Donegal (38%), Mayo (33%); however, Wicklow (44%) and Carlow (75%) are in the southeast. In Mayo and Donegal, in the northwest mountainous coast regions, 7.7% and 4.08% of the population, respectively, live further than a 60-min drive from the nearest specialized service (Table 2, Figure 2).

In Spain, Madrid (97.89%), the northern regions of Asturias (93.35%), Cantabria (94.11%), the Basque Country (87.57%), and Catalonia (84.44%) have high percentages of people living within 30 min from a specialized palliative care service. The same is true for Murcia (92.74%), which is relatively small and has a cluster of palliative care services. On the contrary, the Canary Islands (35%), and the autonomous communities of Castilla y León (38%) and Castilla La Mancha (41%) have a high percentage of people living further than a 30-min drive. Of these, 15.57% of the people in the Canary Islands (331,232 people), 8.02% of the people in Castilla y León (193,274 people), and 7.31% of the people in Castilla la Mancha (148,212 people) live an hour away from the nearest specialized service. In addition, 9.39% of the population of Aragón and 14.17% of the Balearic Islands are not likely to access a service in less than an hour (Table 3).

Within Castilla y León, in the provinces of Soria, Avila, and Palencia, over 80% of their inhabitants live further than a 30-min drive from a specialized service. The same happens for the province of Cuenca, in Castilla la Mancha, and to a lesser extent in Ciudad Real (over 60%). Provinces with higher percentages of inhabitants living further than a 30-minute drive to a specialized service concentrate in the inner land, whereas the majority of provinces at the Cantabrian and Mediterranean coasts tend to have much smaller percentages of population away from palliative care resources (Table 3).

In Switzerland, 22/26 cantons have over 80% of their population within a 30-min drive from a specialized service. Only four cantons have more than 20% of people living further than 30 minutes away from a specialized palliative care service. Three of these are in the mountainous cantons located in the Alps: Valais (21%), Grisons (50%), and Glarus (77%); and one is in the North: Jura (89%). Cantons with a high percentage of people living further than 60 min away are within the same mountainous areas: Grisons (20.27%), Glarus (6.13%), Uri (5.40%), and Valais (3.01%) (Table 4).

## 4. Discussion

Despite an unequal surface coverage of specialized services, the majority of the population in all three countries live within a 0 to 30 min driving time range to a specialized palliative care service: Ireland 84%, Spain 79%, and Switzerland 95%. The percentage of population living in a 30–60 driving time range tends in all cases to be smaller (though in different proportions amongst countries), and those living in a 60–90 min one, little: 0.58% in Ireland, 1.87% in Spain, and 0.51% in Switzerland. However, this means a clear lack of equity between the countries if we look at the number of people living further than 30 min from the closest palliative care service: 9,695,273 people in Spain, 741,950 people in Ireland, and 452,659 people in Switzerland. Of those, 1,508,870 Spaniards, 64,190 Irish, and 68,000 Swiss people live over an hour away. In addition, equity in access is challenged from a sub-national point of view, as within countries large differences can be appreciated. Spanish inhabitants from Castilla La Mancha, Castilla y León, and the Canary Islands live further than a 30-min drive from the nearest service. The northern Irish counties of Monaghan, Leitrim, Donegal, and Mayo; and southern Carlow and Wicklow, are also less likely to reach specialized services in less than 30 min. In Switzerland, the cantons with high percentages of inhabitants living 30 min away from services are Jura, Glarus, Grisons, and Valais.

A number of factors such as the establishment of services in rather populated areas, the ratios of service per population, the population density (or dispersion), the size of the state, or a combination may partially explain these findings. For instance, even though Ireland has a better proportion of services per population, it is also 1.7 times as big in area and as small in population as Switzerland, which means a greater dispersion of the population across the country and a bigger difficulty for specialized services to offer an equitable accessibility to inhabitants. In the Spanish case, the size and the dispersion of services play a role in explaining the variation in the population’s accessibility: The service areas within less than 30 min, which mostly cover the urban regions such as Madrid and Barcelona, do not reach more distant areas. This is stressed by a low ratio of services per regions. Although in Spain, there is no region in which the ratio of people living within 30 min from the nearest service drops below 50%, over half of the regions have under 0.5 services per 100,000 inhabitants.

This study arrives in a moment in which accessibility in palliative care is especially important due to the pandemics, as people with palliative care needs may be more susceptible to the COVID-19 pandemic. Some studies have started analyzing accessibility in context to COVID-19 during the pandemic [57,58,59].

Even though this analysis has provided an understanding of the potential accessibility of these countries’ populations to specialized palliative care services, some limitations need to be acknowledged. Firstly, national directories might have inaccurate or outdated information, or they might have different definitions and quality criteria for accounting specialized palliative care services. In future studies, the “size of the specialized service” can be estimated through figures given in improved national PC directories or the use of a method that applies several sources for their data collection on palliative care services. Although the diverse types of services may have an effect on the estimates, this approach still is meaningful to compare the countries as per the simple aim of calculating distance to the nearest service, independently of the nature of the provider.

Secondly, the data on population that were used do not take into account people with palliative care needs per region but rather the general population estimate potential accessibility: there are other ways of computing the population coverage [60].

Thirdly, the measure of accessibility relies on a simplified model that does not take into account many possible dimensions such as the temporal (time constraints such as the available time and opening hours), the individual component (individual needs, abilities, and opportunities), socio-economic disadvantages, the capacity of the services, the ratio of services to population, or the possibility that there is no bypassing behavior.

In a service area computed by network analysis, there are many parts in between roads that are included in the service area by way of extrapolation. In some cases, this can overestimate the coverage of the service area on further analyses. Some limitations also need to be mentioned in regard to the imperviousness density dataset used for the dasymetric mapping. The dataset considers not only housing areas but also other imperviousness areas such as industrial areas or roads [61]. The population of the small areas is considered to be distributed evenly over the built-up areas, and multi-floor buildings are not considered. This can lead to a misrepresentation of the population distribution.

Lastly, by using services areas through the network dataset via the ArcGIS Online cloud, road network details such as speed impedances, barriers, or road attributes cannot be checked. This makes it difficult to understand what exactly is modeled for the network analysis. The exclusion of temporal parameters in our network analysis, such as the time of year, recurring events such as heavy snowfall or flooding that affect road networks, or other methods of transport different to the car, are also acceptable limitations. However, these do not influence significantly the results in comparison to the strengths of the method due to the scope of the study: a first step in cross-country comparative studies on geographic accessibility to specialized palliative care services.

In our study, it could be argued that dasymetric mapping offers an accurate and more precise estimation of actual residential areas than other methods, especially in comparing countries. Countries may have different types, and differently sized, administrative areas. Dasymetric mapping lessens the significance of the differences between countries. It should be noted that the ancillary datasets used for dasymetric mapping are being improved due to innovations in satellite imagery or the fine tuning of existing datasets. Improved high-resolution population distribution datasets do exist [62,63]. Another strength is that the use of a single network dataset (despite previously cited weaknesses) allows for the use for all the different countries without having to find road networks from different sources. However, using national cadastre data might be the most correct/accurate methodology. Another option could have been the use of crowdsourced network datasets such as OpenStreetMap [64], but crowdsourced data have a risk of incomprehensiveness and low quality [65].

Future accessibility studies comparing different countries may start from this point, by using this method as a reproducible formula, especially in regard to the benefits of using dasymetric mapping in a setting with different countries. Future studies could enhance the accessibility measure by adding other accessibility variables: socio-economic development, age, size of the specialized services, whether these services are private of public, or by adding a service-to-population ratio based on distance, such as the floating catchment method [15,47]. In addition, the use of statistical comparison as well as the contemplate inequality indices such as the GINI (degree of inequality in a distribution of income/wealth) or Herfindahl (measure of market concentration, used to determine market competitivenes) to show to what extent accessibility is unevenly distributed would be desirable.

Replication of this method in other countries or regions would be useful to inform ministries of health, health-care planners, managers, and different stakeholders, though considering that this methodology is new in the specific case of palliative care and represents a first step toward considering accessibility modeling toward palliative care. This model may benefit from testing and additional modeling to be used right away in other countries. The specific characteristics of this health service and the patients they care for still make this accessibility study of particular importance.

## 5. Conclusions

Equity in access to specialized palliative care services differs among and within the countries of Ireland, Spain, and Switzerland: 95% of the Swiss population lives within a 30-minute drive from the nearest specialized palliative care facility, while Ireland and Spain have 84% and 79% of their inhabitants in that driving range, respectively. Citizens from several regions in Spain (Castilla-La Mancha, and Castilla y León), from some Irish counties (Monaghan, Leitrim, Donegal, Mayo, Carlow, and Wicklow), and from some Swiss cantons (Jura, Glarus, Grisons, and Valais), are less likely to access specialized palliative care in a reasonable time.

## Figures and Tables

**Figure 1 ijerph-18-10345-f001:**
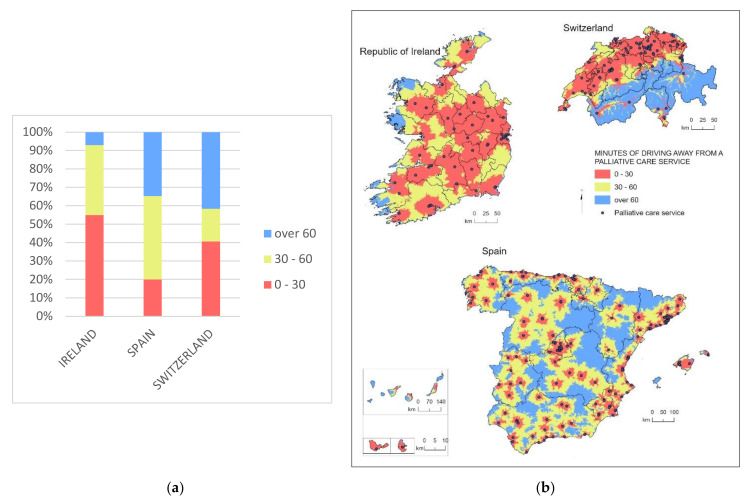
Surface coverage of specialized Palliative Care services (driving time to services). (**a**) Stacked bar chart showing the percentage of surface coverage per distance range in time per country; (**b**) map of the surface coverage of different distance times to the nearest palliative care service.

**Figure 2 ijerph-18-10345-f002:**
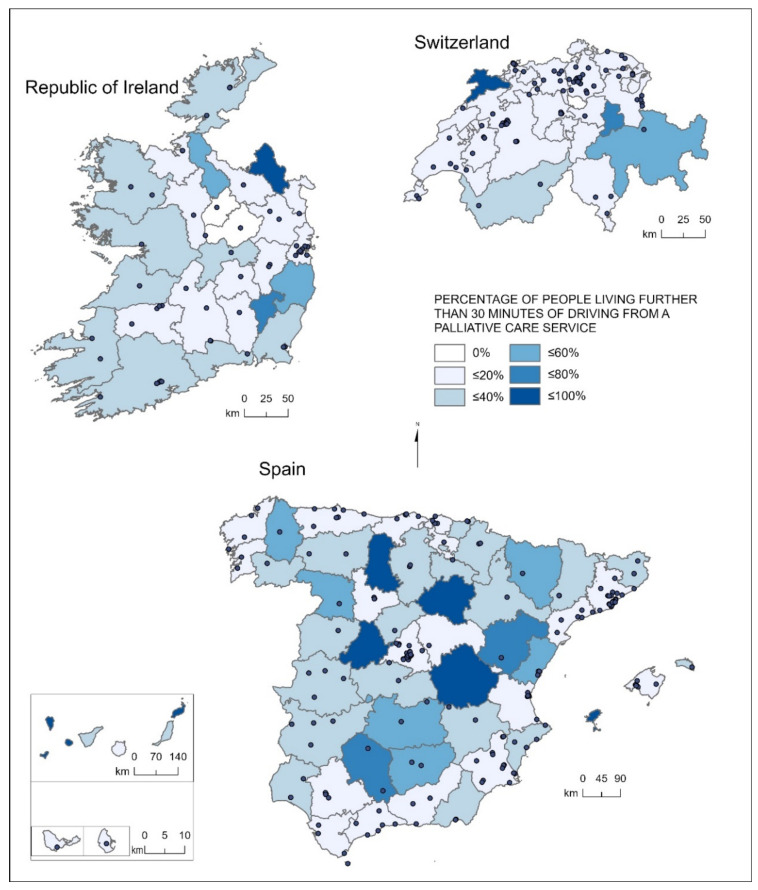
Percentage of people living further than a 30-min drive from a specialized palliative care service in Ireland, Spain, and Switzerland.

**Table 1 ijerph-18-10345-t001:** Countries’ population living at service areas.

Service Areas Driving Time (Minutes)	Ireland		Spain		Switzerland	
	*n*	%	*n*	%	*n*	%
0–10	1,884,201	39.57	17,239,404	36.90	3,941,773	46.46
10–20	1,273,031	26.73	12,966,726	27.75	2,967,508	34.98
20–30	862,683	18.12	6,821,577	14.60	1,122,189	13.23
Subtotal 0–30	4,019,915	84.42	37,027,707	79.25	8,031,471	94.66
30–40	446,579	9.38	4,691,593	10.04	244,503	2.88
40–50	175,881	3.69	2,402,829	5.14	93,006	1.10
50–60	56,899	1.19	1,112,064	2.38	46,955	0.55
Subtotal 30–60	679,360	14.27	8,206,486	17.56	384,464	4.53
60–90	27,414	0.58	876,034	1.87	43,112	0.51
over 90	35,177	0.74	612,753	1.31	25,083	0.30
Subtotal ≥60	62,590	1.31	1,488,787	3.19	68,195	0.81
Grand Total	4,761,865	100.00%	46,722,980	100.00%	8,484,130	100.00%

**Table 2 ijerph-18-10345-t002:** Population living at different driving times from the specialized palliative care services across counties in Ireland.

Counties	0–30 min		30–60 min		Over 60		Total
	*n*	%	*n*	%	*n*	%	*n*
Longford	40,863	99.97	10	0.03	0	0.00	40,873
Westmeath	88,730	99.95	40	0.05	0	0.00	88,770
Meath	193,496	99.21	1248	0.64	300	0.15	195,044
Dublin	1,333,512	98.97	2862	0.21	10,985	0.82	1,347,359
Kildare	218,193	98.06	4311	1.94	0	0.00	222,504
Kilkenny	93,609	94.33	5623	5.67	0	0.00	99,232
Laois	79,071	93.36	5626	6.64	0	0.00	84,697
Limerick	180,987	92.86	13,912	7.14	0	0.00	194,899
Cavan	68,733	90.23	7443	9.77	0	0.00	76,176
Tipperary	143,291	89.81	16,262	10.19	0	0.00	159,553
Louth	115,582	89.68	12,104	9.39	1198	0.93	128,884
Roscommon	54,182	83.95	10,362	16.05	0	0.00	64,544
Sligo	54,058	82.49	10,290	15.70	1186	1.81	65,535
Galway	198,101	76.77	47,708	18.49	12,249	4.75	258,058
Kerry	112,275	76.01	26,448	17.91	8983	6.08	147,707
Offaly	59,015	75.70	18,946	24.30	0	0.00	77,961
Clare	89,232	75.10	28,547	24.03	1038	0.87	118,817
Cork	404,044	74.43	132,674	24.44	6150	1.13	542,868
Wexford	108,421	72.41	39,229	26.20	2072	1.38	149,722
Waterford	84,117	72.41	31,580	27.18	479	0.41	116,176
Mayo	86,514	66.29	34,504	26.44	9489	7.27	130,507
Donegal	98,452	61.84	54,249	34.08	6491	4.08	159,192
Wicklow	80,169	56.29	60,638	42.58	1617	1.14	142,425
Leitrim	16,896	52.73	15,146	47.27	3	0.01	32,044
Carlow	13,769	24.19	43,163	75.81	0	0.00	56,932
Monaghan	4601	7.50	56,434	91.93	351	0.57	61,386
Ireland	4,019,915	84.42	679,360	14.27	62,590	1.31	4,761,865

**Table 3 ijerph-18-10345-t003:** Population living in regions with different driving times from the specialized palliative care services across autonomous regions in Spain.

Autonomous Regions	0–30 min		30–60 min		Over 60		Total
	*n*	%	*n*	%	*n*	%	*n*
Ciudad Autónoma de Melilla	85,010	98.41	0	0.00	1374	1.59	86,384
Ciudad Autónoma de Ceuta	83,399	97.95	0	0.00	1745	2.05	85,144
Comunidad de Madrid	6,438,968	97.89	138,417	2.10	693	0.01	6,578,079
Cantabria	546,052	94.11	27,173	4.68	7004	1.21	579,859
Principado de Asturias	959,820	93.35	59,033	5.74	9391	0.91	1,028,244
Región de Murcia	1,371,193	92.74	105,734	7.15	1582	0.11	1,478,509
País Vasco	1,925,815	87.57	270,928	12.32	2345	0.11	2,199,088
Cataluña	6,417,156	84.44	1,099,479	14.47	83,430	1.10	7,599,736
Islas Baleares	881,288	78.07	87,701	7.77	159,920	14.17	1,128,908
Galicia	2,103,396	77.85	529,309	19.59	69,037	2.56	2,701,743
Extremadura	797,488	74.33	236,666	22.06	38,709	3.61	1,072,863
Comunidad Valencia	3,684,580	74.23	1,192,828	24.03	86,295	1.74	4,963,703
Andalucía	6,138,419	73.21	2,032,086	24.24	213,903	2.55	8,384,408
Aragón	887,355	67.80	298,456	22.81	122,916	9.39	1,308,312
Comunidad Foral de Navarra	436,546	67.41	198,655	30.68	12,353	1.91	647,404
La Rioja	211,313	66.94	98,979	31.35	5382	1.70	315,675
Canarias	1,385,526	65.12	410,927	19.31	331,232	15.57	2,127,685
Castilla y León	1,474,924	61.22	740,966	30.76	193,274	8.02	2,409,164
Castilla-La Mancha	1,199,464	59.18	679,131	33.51	148,212	7.31	2,026,807
Spain	37,027,713	79.25	8,206,264	17.56	1,488,747	3.19	46,721,715

**Table 4 ijerph-18-10345-t004:** Population living in regions with different driving times from the specialized palliative care services across cantons in Switzerland.

Cantons	0–30 min		30–60 min		Over 60		Total
	*n*	%	*n*	%	*n*	%	*n*
Zug	125,421	100.00	0	0.00	0	0.00	125,421
Basel-Stadt	193,908	100.00	0	0.00	0	0.00	193,908
Zürich	1,504,214	99.99	131	0.01	2	0.00	1,504,347
Aargau	670,597	99.94	390	0.06	0	0.00	670,987
Genève	494,638	99.88	613	0.12	0	0.00	495,251
Basel-Landschaft	286,615	99.86	408	0.14	0	0.00	287,023
Thurgau	273,036	99.72	388	0.14	376	0.14	273,800
Fribourg	313,980	99.65	1086	0.34	9	0.00	315,075
Appenzell Ausserhoden	54,738	99.20	440	0.80	0	0.00	55,178
Nidwalden	42,409	98.69	348	0.81	213	0.50	42,970
St. Gallen	493,968	97.88	10,167	2.01	552	0.11	504,687
Vaud	773,376	97.51	19,584	2.47	168	0.02	793,128
Luzern	393,541	96.81	12,960	3.19	0	0.00	406,501
Solothurn	261,258	96.25	10,173	3.75	0	0.00	271,431
Appenzell Innerrhoden	15,479	96.11	626	3.89	0	0.00	16,105
Neuchâtel	169,447	95.21	8517	4.79	0	0.00	177,964
Schwyz	148,908	94.66	6238	3.97	2157	1.37	157,303
Ticino	334,214	94.49	15,007	4.24	4487	1.27	353,708
Bern	948,478	91.99	78,635	7.63	4008	0.39	1,031,121
Schaffhausen	70,232	86.33	11,121	13.67	0	0.00	81,353
Obwalden	31,750	84.50	5394	14.36	430	1.14	37,574
Uri	29,407	81.01	4931	13.58	1961	5.40	36,299
Valais	267,645	78.38	63,560	18.61	10,262	3.01	341,467
Graubünden	117,067	59.16	40,717	20.58	40,102	20.27	197,886
Glarus	9128	22.62	28,747	71.24	2475	6.13	40,350
Jura	8016	10.94	64,283	87.71	993	1.35	73,292
Switzerland	8,031,470	94.66	384,464	4.53	68,195	0.81	8,484,129

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
