# Peer review of "Population’s Potential Accessibility to Specialized Palliative Care Services: A Comparative Study in Three European Countries"

_ijerph, 2021, doi:10.3390/ijerph181910345_

Round 1

Reviewer 1 Report

Major Comments

  1. Only measure travel distance does not help to accurately measure the spatial accessibility to the palliative care service. As you mentioned in line 45 – 47. The accessibility measurement comes from the interactions between supply, demand, and traffic. Here, the supply would be the number of physicians in the palliative care; the demand would be the number of people who need palliative care service; the traffic would be the travel distance. You may wish to refer the following papers:

Luo, W., & Wang, F. (2003). Measures of spatial accessibility to health care in a GIS environment: synthesis and a case study in the Chicago region. Environment and Planning B: Planning and Design, 30(6), 865-884

Luo, W., & Qi, Y. (2009). An enhanced two-step floating catchment area (E2SFCA) method for measuring spatial accessibility to primary care physicians. Health & place, 15(4), 1100-1107.

  1. What is the justification of 30-minute intervals?
  2. I don’t think it is appropriate to use the general population in your analysis. It’s not true that general population need palliative care services. Palliative care services may be placed for the people who have very serious illness. In this sense, you may wish to adjust the population in your analysis.
  3. If you wish to compare the spatial accessibility, you may need to provide the statistical comparison at least (e.g., ANOVA or something). Also, as you mentioned an inequity in the introduction, you may need to calculate the Gini or Herfindahl indices to show to what extent each country’s accessibility is unevenly distributed.
  4. Based on the comparison in your study, you may also be able to draw conclusions regard to the current COVID-19 situation. People staying at palliative cares may be more susceptible to the COVID-19, and some studies measure spatial accessibility in the context of the COVID-19.

Tao, R., Downs, J., Beckie, T. M., Chen, Y., & McNelley, W. (2020). Examining spatial accessibility to COVID-19 testing sites in Florida. Annals of GIS26(4), 319-327.

Ghorbanzadeh, M., Kim, K., Ozguven, E. E., & Horner, M. W. (2021). Spatial accessibility assessment of COVID-19 patients to healthcare facilities: A case study of Florida. Travel Behaviour and Society24, 95-101.

Kang, J. Y., Michels, A., Lyu, F., Wang, S., Agbodo, N., Freeman, V. L., & Wang, S. (2020). Rapidly measuring spatial accessibility of COVID-19 healthcare resources: a case study of Illinois, USA. International journal of health geographics19(1), 1-17.

Minor Comments

  1. You may wish to provide the full name when the abbreviation appears first. (e.g., the WHO) (Line 31)

Reviewer 2 Report

Review: 

This article is a contribution to the recently large literature on physical accessibility to healthcare, focusing on palliative care services. It is relatively well written. However, I have several concerns related to the ways the analyses have been done, and the extent to which this type of analysis can be replicated, as is, in order compare overall accessibility between countries. In general, the authors must present better the assumptions and the details behind their methodology, and must relate to the large recent body of literature that already exists on access to health services, as well as population distribution estimates. 

I have several suggestions for major changes, followed by some minor ones:

Major:

- lines 47-56: It is indeed important to give to the readers a good overview of how accessibility modeling has been used to better understand/plan health systems. However, your treatment of this here is too short, and it's not clear whether you seek to cover the approaches used only in high-income countries, or (as it seems) approaches applied also in low and middle-income countries (LMIC). My main concern is that you are using mainly quite old references. There has been a lot of recent work in this area, and it would be beneficial to the reader that you better place your study within the recent body of work. Here are a few recent publications (and see references within) to consider, as a start:

  • Macharia PM, Ray N, Giorgi E, Okiro EA & RW Snow. 2021. Defining service catchment areas in low resource settings. BMJ Global Health, 6: e006381 https://gh.bmj.com/content/6/7/e006381.full
  • Ouma P., Macharia P.M., Okiro E., Alegana V. (2021) Methods of Measuring Spatial Accessibility to Health Care in Uganda. In: Makanga P.T. (eds) Practicing Health Geography. Global Perspectives on Health Geography. Springer, Cham. https://doi.org/10.1007/978-3-030-63471-1_6
  • Paez, A., C. D. Higgins and S. F. Vivona (2019). Demand and level of service inflation in Floating Catchment Area (FCA) methods. PLOS ONE 14(6): e0218773. https://journals.plos.org/plosone/article?id=10.1371/journal.pone.0218773
  • Ebener S, Stenberg K, Brun M, Monet J-P, Ray N, Sobel H, Roos N, Gault P, Morrisey Conlon C, Bailey P, Moran AC, Ouedraogo L, Kitong J, Ko E, Sanon D, Jega FM, Azogu O, Ouedraogo B, Osakwe C, Chimwemwe Chanza H, Steffen M, Ben Hamadi I, Tib H, Haj Asaad A & T Tan Torres. 2019. Proposing standardised geographical indicators of physical access to Emergency Obstetric and Newborn Care in low- and middle-income countries. BMJ Global Health4: e000778 https://gh.bmj.com/content/4/Suppl_5/e000778

Your methodology in itself is not innovative (and may suffer some drawbacks, see below), but if accessibility modeling has really not been done to compare countries in the framework of palliative care, you should state it more clearly and better describe the specificity of this particular health service compared to other health services.

- Lines 126-139: this catchment model is at the heart of your methodology and results, and it's where you may receive the most critics in light of the large body of work on realistic accessibility model that is being done in other countries. Indeed, there is not enough information on the data you used from ArcGIS online:

  • What is the source of the road data ArcGIS is using in each of the three countries?
  • Are all the official road categories being considered?
  • Are the official maximum driving speed being used on each road category, in each considered country ?
  • You mentioned the use of "traffic lights", is the modelled travel time considering waiting time at traffic light, or any other traffic related information? If so, is it averaging travel speeds/times at different times of the day/week/year ?

The reader needs this information (potentially in an Annex) to understand what has been modelled. You need to convince the reader this is the most realistic (or at least realistic enough for your aim) way to compute travel time and catchment for these three countries, and better present its limitations. This brings me to the next issue.

The Network Analyst is great to compute travel time in areas with dense roads, because the catchments areas generated will be relatively realistic when it comes to what is happening in between the roads. The way Network Analyst works is that it extrapolates what is between the roads to be part of the catchments. In certain area where the road network is not that dense, and where the population may use other means of transport to reach the nearest road (walking, motorbike, etc, as is the case in many LMIC countries), this can greatly overestimate population coverage. This should be part of the limitations in the Discussion part.

- Lines 141-151 population

Considering dasymmetric methodology to downscale census data in a building footprint is the right way to do it, as exemplified in man recent publications about access to various services, e.g. see Giuliani et al, 2021 (https://www.mdpi.com/2072-4292/13/3/422/htm ) who modelled access to urban parks in Switzerland. However, I wonder why you haven’t used the ready-to-use products from WorldPop (https://www.worldpop.org/geodata/listing?id=79 ) that are using advanced dasymmetric technique, a very recent building footprint, and a set of authoritative covariate data sets in the process. 100-m resolution data sets are available for your three countries from Worldpop. If there is a specific reason why you have used your methodology instead, you must explain. I encourage you to use WorldPop, but if you keep your methodology, you must add much more details (in a Supplementary?) about the way you conducted your dasymmetric modelling. It was not clear to me what is this "imperviousness density", and how exactly you used it. What spatial raster resolution was used? Also, how did you distribute the population in each building? Did you assume the same number of people in each building? Did you account for multi-floor building? Did you consider you captured well the residential buildings, or were there potentially numerous nonresidential buildings considered ? You need to describe your assumptions for this process.

- Lines 261-278, Limitations: this part on limitation should be extended with the following items: (1) in the Introduction you introduced the fact that there are many types of palliative care, hence many possible ways/incentives/care seeking behaviours to reach them, but you are not discriminating between these different types in your analyses. What could be the effect of that on your estimates? What if the difference between palliative care types is very different between countries, is it really meaningful to compare the countries with your approach ?, (2) you are assuming there is no by-passing behaviour (i.e. not going to the closest palliative care service), this should be stated somewhere, (3) you have not captured any uncertainty in the way you compute the population coverages (as in, for example, Curtis et al, 2021 https://bmjopen.bmj.com/content/11/7/e045891.full ). This should be acknowledged in the limitation, especially because you insisted that this methodology could be used to compare countries, which calls for caution when interpreting any observed differences among countries (or admin units at sub-country level). As you probably know, changing the travel speeds a little bit could change your population coverages and therefore the way the differences are interpreted among countries.

- Line 275.-278: I would rewrite that part. The exclusion of temporal parameters should not be seen as "less relevant", it's rather that you are assuming that not accounting for those does not influence significantly your results. This is a limitation.

- Lines 279-289:  I would rewrite that paragraph by toning down your saying on dasymmetric mapping. There are a lot of papers (see e.g. the papers from the WorldPop team, from Andy Tatem and colleagues) that already discuss in length this type of approach compared to other approach for considering population counts from census. Also, in European countries OSM data are relatively good, but the best data sets are the official road layers from national cadasters. It is available from Switzerland upon request to the central administration, and might be as well in Spain and Ireland (I don't know the situation there). Going forward, and before there is a high-quality road data sets (with associated driving speeds) freely available at European level, it might be that using cadaster data might be the most correct/accurate methodology if one really wants to be able to compare countries.

Lines 290-298: I would also tone down and rewrite the conclusive part of your Discussion. The question the reader will ask is: if I'm a decision-maker/stakeholder in any of these countries, can I take the results of this study and use them to fully assess or improve my palliative care network ?
If your methodology is a first step towards considering accessibility modeling towards palliative care, there are several issues (that I mentioned above) that need to be addressed first, with stronger arguments (and possibly testing and additional modelling) to be able to recommend that your methodology can be used right away, as is, in other countries. Also, you are saying that future studies could enhance accessibility measure by adding components. You could add a bit more on how they could do that and to what purpose. For example, it would be interesting to know how you would add "size of the specialized services", or the "service-to-population ratio" to your methodology (can you really mix travel time-based measure with these ratio estimates?).

Minor:

- line 33: change to "… a global resolution which urged …"

- line 113: change to "With its location partly in the Alps, …"

- line 116: "… population live in only four of them."

- line 137: "… were considered reasonably differentiated …"

- Figure1: I suggest removing the blurry borders around each figure panel

- Figure 1 legend: remove the last sentence in the legend.

- Line 215: There is no Figure 3, so I guess you meant Table 3 (or Figure 2) ?

- overall: I suggest that you keep the same order for the three countries (e.g. Spain, Ireland, Switzerland) when you use them in a sentence, especially in the Results and Discussion section, this will ease the reading for the reader.

- Line 241: change "huge" by "large"

Round 2

Reviewer 2 Report

You have addressed my comments, thank you.